# Determinants of Smart City Commitment among Citizens from a Middle City in Argentina

María Verónica Alderete 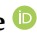

IIESS, CONICET-UNS, Department of Economics, UNS, Bahía Blanca CP 8000, Argentina; mvalderete@iiess-conicet.gob.ar; Tel.: +54-2914595138

**Abstract:** This paper aims to examine the determinants of smart-city commitment across individuals from Bahía Blanca, Argentina. Literature has identified different factors explaining citizens' commitment to smart cities, such as education, age, labor condition, and other more subjective factors, such as trust and awareness about the smart-city concept. A mediator factor of smart commitment is e-readiness or digital readiness, that is, the level of preparedness to properly exploit internet opportunities such as e-government and e-commerce. To achieve this goal, we used a survey conducted on 97 citizens (followers of the Moderniza Bahía Facebook) from the city of Bahía Blanca, Argentina. By estimating a structural equation model, we found that higher levels of ICT use are associated with higher levels of smart-city commitment and that higher awareness of the smart-city concept is related to higher levels of smart-city commitment. Sociodemographic factors such as age and labor condition also explain ICT use.

**Keywords:** smart cities; ICT use; sociodemographic factors; e-readiness





## 1. Introduction

Smart cities are a new style of a city designed to encourage healthy economic activities with the help of information and communication technology (ICT) while improving quality of life and providing sustainable growth. We are witnessing a more interconnected and challenging context that imposes the need to find better solutions for information sharing and transfer among multiple actors involved in shared environments, such as urban cities [1].

Under this smart-city scenario, information and communication technology infrastructures are critical tools to improve and support social and urban growth, promoting citizens' participation and governmental efficiency. From the citizen perspective, the adoption of ICT is critical for the performance of smart cities and smart activities. Smart-city services offer citizens a better living environment and raise their quality of life [2–4]. There are many examples of smart-city services such as end-to-end applications that focus on common concerns such as mobility, public services and safety. Recently, the crucial role of these smart-city services has been evidenced because of the pandemic crisis [5].

Since urban services are used by citizens, it is of main importance to take account of their opinions and perceptions for correct management of the services [6]. Nowadays, cities do not have a deep enough understanding of their citizens to actively and effectively engage them [7] (p. 12). Scholars have highlighted that little research has focused on examining the citizens' commitment to smart cities such as smart practices or activities [8]. The authors believe there is a gap between the recent political approach that focuses on citizens' "engagement" or "participation" in smart cities and research related to this issue. In fact, the smart city differs from the digital city in the focus given to the relationships among the actors and entities involved and their contribution to the success of all [9]. However, some recent empirical literature has explained the factors that motivate citizens' engagement in smart cities [6,10,11]).

The framework of this study is in line with the diffusion of innovation theory and incorporates the claims of the unified theory of acceptance and use of technology (UTAUT) [12]. Based on the UTAUT, citizens (users) with different socioeconomic characteristics act differently regarding the usage of technology for performing smart activities. While most of the empirical literature on smart cities focuses on the supply side, which is the role of firms providing smart services, government policies for building smart cities and urban planning, among others, there is scarce research on the demand side from the citizens' perspective. This gap is especially pronounced in the context of developing countries.

Empirical literature on smart-city commitment in cities of developing countries is still scarce [13,14]. Thus, this paper contributes to the expansion of research on this subject. Building on this reflection, this study provides some understanding of the factors that are related to the success or failure of citizens' engagement with smart-city initiatives through an explorative study on the smart-city landscape. Does ICT use mediate the relationship between smart-city commitment and sociodemographic factors? In order to accomplish this goal, a structural equation model (SEM) was estimated by employing an online survey of a group of citizens from Bahía Blanca.

We examined the city of Bahía Blanca because it is one of the most transparent cities in Argentina. Based on the Open Data Index (ODI) of the Open Knowledge Foundation, the city is ranked fourth at the top of the ranking. During the last years, the city has changed the relationship between local government and citizens through the adoption of innovative actions and projects, a more transparent government, the opening of data, the implementation of practices to promote citizens' engagement and the adoption of new technologies [15].

Bahía Blanca is a city of about 400 thousand inhabitants located in the southwest of Buenos Aires Province, Argentina. Bahía Blanca is also one of the Argentinean cities awarded for the program "Pais Digital". Based on the latest data on ICT diffusion at the household level, Bahía Blanca is one of five urban places in Argentina with the highest percentage of households using the Internet [16]. In Argentina, 69.4% of individuals use the Internet (two out of three people use the Internet), but only 16.1% of inhabitants have a fixed broadband subscription. On the other hand, active mobile broadband subscription is higher than fixed broadband reaching 67% [17].

Better exploitation of Internet opportunities depends on the education level of the population and their digital skills. In this aspect, data based on EPH (Household Permanent Survey) 2016 show that 34% of people older than 25 hold a tertiary or university level complete in the Bahía Blanca Cerri conglomerate. Moreover, 60% of people older than 20 have at least the secondary school level complete.

The paper is structured as follows. First, we describe the theoretical framework related to smart cities necessary to establish the hypotheses of the study. Secondly, the methodology and data are explained. Thirdly, we show the results obtained by estimating the structural model, and finally, results and conclusions are provided.

## 2. Theoretical Framework

During the last years, some research has emerged on the importance of building smart cities. Although the concept is not new and there are plenty of definitions, we can distinguish two different approaches. On the one hand, a technological view centers on the ICT role as a means to deepen and strengthen access to public information and to make city services more efficient [9,18–21]. On the other hand, a wider approach is related to notions of sustainable economic growth, quality of life, participatory governance and the reduction of $CO_2$ emissions that place the citizen at the center of analysis [19,22–26]. According to this view, "the spirit of e-governance in a smart city should be citizen-centric and citizen-driven [22] (p. 12)". Most definitions of smart cities in this approach coincide in five features or characteristics: smart economy, smart governance, smart mobility, smart environment and smart people.

At the same time, the evaluation of citizens' perceptions with respect to urban innovations has become relevant [27,28]. Macke [27] focused on a wider definition of smart cities and evaluated the perception of the quality of life of Curitiba, Brazil. From a theoretical perspective, the diffusion of innovation theory (DOI) [29] offers a model to explain the social and technological factors that affect the acceptance of innovative information technology (IT). This theory is broadly employed for comparison of initial adoption and continued use in a variety of technologies [30]. Later on, Moore and Benbasat [31] proposed the innovation diffusion theory (IDT), which is an extension of the DOI theory, but it also includes in ICT adoption a moderating factor termed "personal innovativeness". This term represents an individual's propensity to innovate [30]. If users are more novelty-seeking, they will accept more innovative technologies. IDT focuses on the technological aspects and then incorporates the users' behavior and social factors associated with the innovative technology.

Recent empirical evidence has analyzed the determinants of citizens' engagement in smart-city initiatives. For instance, Novo Vázquez and Vicente [32] analyzed the factors that outline citizens' e-participation in smart cities. The authors found that political engagement, ICT usage and socioeconomic factors, such as age, educational attainment and labor situation, explain participation.

In addition, Cardullo and Kitchin [11] analyzed the role and context of citizens in smart cities by examining smart-city initiatives in Dublin, Ireland, which they titled "the scaffold of smart citizen participation". On the other hand, Polese et al. [10] offered an empirical investigation from a sociological and psychological perspective on the perception of citizen services in the city of Brno, the Czech Republic. The study centered on the smart-cities issue as an example of complex service systems (CSSs) and showed how actors' perceptions (cognitive and psychological dimensions) can influence opportunities and willingness for value co-creation and collaboration.

Similarly, Caputo et al. [33,34] analyzed the role of individual perceptions in the evaluation of services and in the definition of possible collaborative behaviors, with specific reference to the domain of the smart city. On the other hand, Yeh [6] examined citizens from Taiwanese cities that had participated in smart-cities campaigns. Citizens will use smart-cities services if they are innovative, high-quality and secure (in terms of privacy).

Moreover, dimensions utilized to evaluate the performance of cities range from the most technological ones to aspects related to citizen participation and engagement. Most smart-city indexes (i.e., Smart City Index, IESE Cities in Motion) are based on six dimensions: economy, people, governance, mobility, environment and life. Each of these dimensions is decomposed into elements or factors and each factor into a set of indicators [35].

## 2.1. SC Activities or SC Commitment

Since the increasing focus of smart cities on citizens, there is a need to enable citizen engagement by fostering participation, collaboration and community empowerment [36]. In this study, we supposed that citizens' engagement with the smart city should be reflected in the accomplishment of many smart activities such as e-government, e-commerce and care for the environment, among other dimensions.

The smart economy dimension is critical to measure the economic "health" of a smart city [37]. This aspect is related, directly or indirectly, to smart and ICT infrastructure for connectivity. In addition, it refers to "soft" capabilities such as smart health and smart funding [37]. Businessmen are usually the first adopters and drivers of ICT in the economy. In this paper, we focus on the citizens' ICT adoption for buying and selling online. By participating in e-commerce, citizens help to foster a competitive economy, where competition and competitiveness take place [38].

The smart people dimension stresses that human and social capital is an important tool to develop smart cities. Prosperity in any smart city involves investment in human capital and related infrastructure [39]. This explains why knowledge workers are frequently apt to concentrate around cities [4]. Technological devices and digital resources are useful tools to

enhance people's lives if and only if their users are ready and prepared to properly utilize them [21,40,41]. In smart-city frames, the demand for smart learning has increased steadily due to the integration of digital technologies and the Internet into learning [42].

The smart governance dimension characterizes the efficiency and qualification of the state intervention to solve citizens' requests. Governance is associated with e-government practices, mainly citizens' engagement [43]. In many countries, e-government initiatives are common because they imply more citizen-centric governance [44]. E-government conveys new ideas such as transparency, accountability and citizen participation in the evaluation of government performance under this knowledge-society approach [45,46]. Moreover, processes of political and electronic participation are a baseline in the development of smart cities [47,48]. Citizens perform smart practices if they realize the potential economic and environmental gains of living smart [49]. On the other hand, Granier and Kudos [8] explained the case of Japanese citizens and found that smart communities' goal is to encourage citizens to participate in the co-production of public services but not involve them in governance.

The smart environment dimension implies citizens developing sustainable and scalable practices such as classification and/or recycling of waste, using less energy and purchasing second-hand goods, among other environmentally friendly practices [37]. While the built environment (including both buildings and supporting infrastructure) will need to support the rapid increase of population and urbanization, cities will have to adapt to address climate change and its associated impacts. Many studies show that citizens from smart cities have a strong commitment to the development of sustainable and scalable practices such as trash recycling and efficient use of energy resources, among others [8,41,50–52].

From an environmental perspective, Sovacool [53] suggested that citizens' participation increases democracy, especially if citizens perceive they are represented in environmental decision making. In addition, public engagement and learning lead to behavior change related to environmental issues [54,55]. Bull and Azennoud [54] offered an example of citizen engagement in Hampshire, USA, where a discussion of an adequate waste strategy was performed to engage the public in decision management of household waste in the city.

Finally, smart mobility also plays a key role in promoting the development of smart cities. In metropolitan large areas, mobility is considered one of the most challenging topics to confront. It comprises both environmental and economic factors and needs both high technologies and smart people. Smart mobility is mostly embedded in ICT to support optimum traffic and to collect citizens' opinions regarding livability in cities or quality of local public transport services [56].

### 2.2. ICT Use and SC Activities

Technology trends in smart cities such as mobile broadband connectivity, open data, urban interfaces and cloud computing, among others, are progressing at a rapid pace. ICT plays a key role in smart cities by connecting infrastructures, government and citizens [7]. Paskaleva [21] highlighted that a smart city is based on the "use" of advantages offered by information and communication technology.

By using ICT, public services can be offered online, public information can be easily accessed and citizens can organize among themselves to share interests and concerns. Moreover, ICT use can facilitate and promote citizens' decision making and involvement in public life [32].

ICT has an increasing impact on different aspects of citizens' quality of life [6,57–60]. For instance, the adoption of ICT applications in e-government is a prerequisite for the emergence of smart cities [14]. By using e-government, citizens can communicate with governments from different levels (national, federal, regional, local), and, by being involved in public-sector governance, they improve its efficiency and effectiveness. In this vein, Lytras and Serban [61] established that one of the most critical applications of smart cities in contemporary societies is e-government. However, "although smart city services are

driven by advanced information technologies, their success is highly dependent on user engagement, which is historically problematic [49] (p. 845)".

Socioeconomic demographics differences can give rise to a digital infrastructure gap between cities in emerging economies, such as India, asking for smart-city policies implementation [62]. However, in some countries such as India, most technologically advanced cities such as Ahmedabad had only 10.3% of its households with Internet access in the year 2011 [62]. Therefore, ICT access has not yet been considered a basic infrastructure for building smart cities in some less developed countries.

ICT use depends not only on connectivity (Internet access) and access to digital devices such as computers, tablets and mobile phones, among others, but also on ICT skills. This relationship between access and use is due to the existence of complementarities in ICT diffusion [63]. Personal computers have a stronger impact on Internet diffusion in developing countries than in developed countries. Usually, unequal access to resources and rights is correlated with both ICT appropriation and ICT intensity use. Then, we can state the first hypothesis:

**Hypothesis 1 (H1).** *ICT use positively affects citizens' performance of SC activities.*

### 2.3. Awareness of the SC Concept and SC Activities

Studies based on the IDT have shown that the perceived characteristics of innovation can have a significant effect on adopters' behaviors toward the acceptance and usage of innovative ICT solutions [64], e-government systems [65–67] and SC services [6]. Similarly, awareness of the smart-city concept is a main factor to explain smart citizens' behavior. However, Neirotti et al. [48] found a negative relationship between awareness and SC activities through a sample of 70 international cities. A possible explanation is that the most polluted cities are located in developing countries in which complete awareness of the SC concept has not yet been established.

**Hypothesis 2 (H2).** *The awareness of the SC concept positively affects citizens' performance of SC activities.*

### 2.4. Trust and SC Activities

Trust is generally understood as the willingness of one person to count on the behaviors of others, especially when this person is in a vulnerable situation. Trust always supposes the person is ready to accept a certain degree of risk and to become vulnerable to a trusted party [68]. This approval of risk is based on the expectation that the trusted party will perform actions that are important or beneficial to the vulnerable party [69].

If citizens trust in ICT-based SC services, they will positively accept and use the ICT-based SC services [6]. However, Novo Vázquez and Vicente [32] did not find trust in governments to be a significant factor explaining e-participation, which is one of the dimensions of smart cities (smart governance). Therefore, we can pose hypothesis 3.

**Hypothesis 3 (H3).** *Citizens' trust in e-government positively affects citizens' performance of SC activities.*

### 2.5. Sociodemographic Factors and SC Activities

Demographic characteristics affect the use of a given product or service [70], probably because the needs of citizens vary according to their educational level, gender and age. However, results on the impact of demographic factors on usage are not conclusive. Yeh [6] showed that Taiwanese citizens' attitudes and behaviors toward ICT-based SC services are not affected by their demographic characteristics in terms of age, gender and education reinforcing the public nature of services, which are oriented toward the whole community. On the other hand, Novo Vázquez and Vicente [32] observed that socioeconomic factors such as age, educational attainment and labor situation are significant factors explaining citizens' e-participation, while gender is not significant.

Citizens' education is one of the main determinants of people's digitalization level [71]. The educational condition is a significant predictor of ICT use. An educated individual can access cognitive resources, digital skills and, more importantly, social and knowledge resources. Some studies have claimed that education affects the probability of using the Internet [72] and the diffusion of the Internet [73,74]. In particular, Hargittai [74] observed that education significantly enhanced the penetration level of Internet hosts across countries. In addition, Kiiski and Pohjola [73] showed that mean years of schooling is a significant factor explaining the growth of Internet hosts per capita. On the other hand, Chinn and Fairlie [75] did not observe a significant correlation between the variables.

Hence, education can indirectly determine the development of SC activities. In this line, Belanche, Casaló and Orús [52] showed that citizens' educational level positively affects urban services usage since the more educated the people, the higher the environmental consciousness will be. The higher the educational level (human capital), the greater the probability of using ICT will be. People that are more educated have better access to information about SC benefits and how to perform intelligent activities. Hence, the development of SC activities will be higher.

**Hypothesis 4 (H4).** *The more educated the people, the higher the ICT use will be, and therefore, the higher the citizens' performance of SC activities might be.*

Gender is also considered an important factor to predict the use and acceptance of technology-related applications or systems. Decision-making processes are different between men and women [76–78]. Tarhini, Hone and Liu [77] found that gender was not significant in influencing the relationship between perceived ease of use and behavioral intention of students' e-learning. Moreover, Al-Shafi and Weerakkody [79] observed that gender significantly moderates the effect of the determinants on behavior intention to use the wireless Internet park service in Qatar. In particular, males were positively associated with higher intention to use. On the contrary, Mensah [76] did not find gender to be a significant demographic factor in explaining the willingness to use e-government services.

**Hypothesis 5 (H5).** *Men are more likely to use ICT than women are; therefore, they are more likely to perform more SC activities than women are.*

On the other hand, older people perceive lower potential profits from innovating than younger people because they attribute a higher discount rate to future profits. While young people are more likely to have ICT access [80], older people usually use the Internet to a lesser extent [71,81]. There are differential effects regarding the affect–creativity association, depending on employees' age [82]. Therefore, age is a proxy of personal innovativeness and can be employed as a preliminary factor to understand the characteristics of adopters [30] and the level of acceptance/usage of ICT-based SC services [6].

**Hypothesis 6 (H6).** *Older people are less likely to use ICT and, therefore, to perform SC activities mediated by ICT such as e-commerce and e-government, among others.*

Finally, income is an important determinant of ICT adoption as it indicates the households' budget constraints [72,80,81]. One of the users' characteristics that affect the intra-country digital divide is related to income [83]. The greater the household income, the higher the probability of using the Internet is [84]. The lack of income does not allow citizens to take part in the ICT demand. Therefore, unemployed or people with a transitory job are less likely to use ICT.

**Hypothesis 7 (H7).** *Citizens with a permanent job are more likely to use ICT and, therefore, to perform SC activities.*

Based on this research background, the model examines the causal relationship between smart-city activities and sociodemographic factors by describing the mediating role of ICT use. Hence, citizens' gender, age, education level and labor condition are the control variables of the proposed framework by explaining directly ICT use and indirectly SC activities. Figure 1 offers a graphical representation to explain the model.

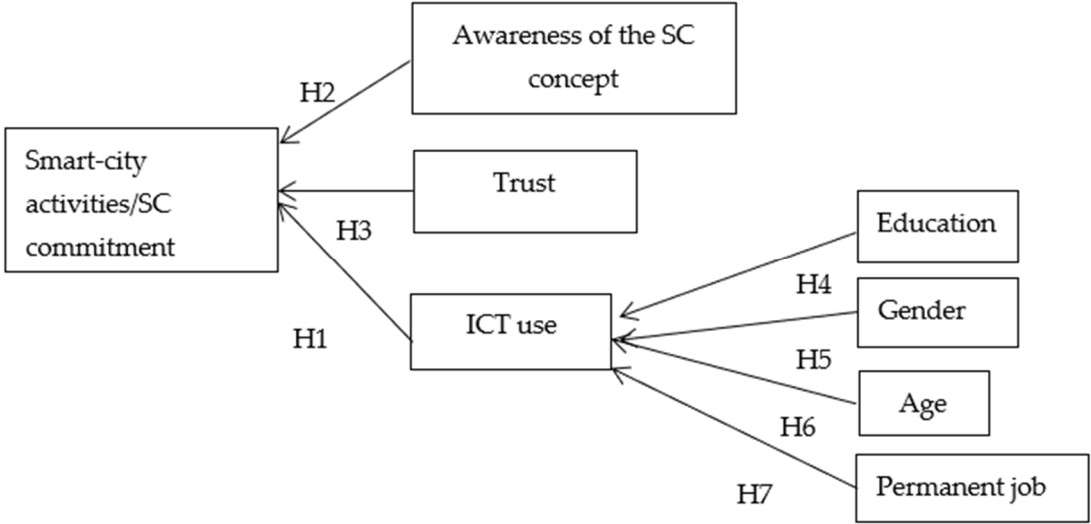

**Figure 1.** The model to be estimated Source: Own elaboration.

## 3. Methodology

Bahía Blanca is a city of around 400 thousand inhabitants situated in the southwest of Buenos Aires Province, Argentina. According to the National Survey on ICT for the year 2011 [85], 53.8% of urban inhabitants had fixed Internet access and 43.9% had a personal computer in Bahía Blanca. These percentages were higher than the national level of Internet access (38.8 %).

Since it would have been costly to obtain a representative sample of the whole population of the city, we decided to limit the population under study to the followers of the Facebook fan page "Moderniza Bahía". Nowadays, this page has the name of Lab Bahía, a place to promote citizen participation and collective intelligence to bring innovative solutions to local problems. The lab is part of the local government initiative, and it is very active in terms of open government and open data. The number of followers is stable, with around 17 thousand followers. The type of interaction is mainly unidirectional. A couple of recent publications with many comments and shares have concerned the transformation of the city park "Parque de Mayo" and building of the app "Organiza tu salida" to manage walking around because of COVID-19.

We used data obtained from an online survey conducted between April and May 2019, which was supported by the Secretary of Modernization of Bahía Blanca Town Hall. During these two months, we collected data from 97 citizens. The original sample (N = 99) included two invalid cases of individuals who did not live in the city. As a result, there were too many missing data in those cases. These two cases were not included in the analysis. Besides, the sample is biased and does not represent the overall population of Bahía Blanca. Therefore, results obtained are only representative of follower citizens of that social media website.

The questionnaire was based on several literature sources on e-government and smart cities [86–88]. It collected information about the citizens' profile (age, education, labor condition and ICT use), smart cities and e-government. Questions were closed, and most of them were Likert-scaled.

The size of the sample was 97; we collected 97 valid observations during the two-month period of diffusion of the survey. The sample size was enough for a finite population sampling (lower than or equal to 100,000) [89]. The sample responded to the following formula:

$$n = \frac{Z^2 * N * P(1 - P)}{(N - 1) * K^2 + Z^2 * P(1 - P)} \tag{1}$$

where *N* is total population. In this case, followers of Moderniza Bahía Facebook, *N* = 16,400 by April 2019.

*Z* is the value obtained from significance levels. If population distribution is normal at a significance level of 95%, z value is 1.96. In this case, with a significance level of 90%, the corresponding z is 1.645, and so on.

*K* is the error or maximum difference between the sample proportion and the population proportion to be accepted based on the significance level proposed. For a significance level of 95%, k is 0.05; similarly, for a significance level of 90%, k is 0.1, as in this case.

*P* is the population fraction of interest, a parameter that indicates the percentage of population interested in smart cities. It can be obtained from a pilot survey, but if it is an unknown parameter, as in this case, it is suggested to use the most unfavorable value of 50%.

For each question, respondents were asked to declare their opinion using a five-point Likert scale [90] in which 1 = "strongly disagree" and 5 = "strongly agree".

In a brief characterization of the sample, we found that 69% were female (the majority of the sample) and 56% were young people between 22 and 40 years old (followed by 37.6% of adults between 41 and 60). On average, the education level of the sample was high, with 38.2% with a university degree and 20.6% with postgraduate studies. On the other hand, 79% had a permanent job. Almost the whole sample had Internet access at home and used mobile phones. The profile of the participants was congruent with the literature on the subject: citizens who regularly use the Internet and are familiar with the digital world, mainly use smartphones and are already aware of the smart-city concept and related terms [91–93].

In terms of the smart-city concept, 42.3% had heard about the concept but did not know enough, and only 2% had worked or implemented smart-city initiatives. Moreover, 25% of the sample did not connect with the local government through the website. Thus, 25% were not using e-government, and only 6% of the sample achieved a transactional level (download, share or re-use databases). On the other hand, 67% connected to Internet to buy and sell products online (e-commerce), and 41% of the sample had recycled or classified waste.

We studied the relationship between smart-city activities and ICT use by estimating a structural equation model (SEM). Although there are different statistical packages, we utilized STATA 14 to perform the estimations. STATA software only uses CB-SEM [94].

In this frame, SEMs are of two different types with regard to the statistical approach (non-parametric or parametric testing), the objective of the study (exploratory or confirmatory) and, especially, the algorithm employed (generalized least squares (GLS) or maximum likelihood estimator (MLE)) (Esposito, 2009). These differences lead to variance-based structural equation modeling (VB-SEM) and covariance-based structural equation modeling (CB-SEM), respectively.

This paper uses a CB-SEM model, which is SEM based on the covariance matrix. CB-SEM goal is to test and confirm the theory by using data (exploratory model). The difference between the theoretical covariance matrix and the estimated covariance matrix (based on data) is minimized to estimate the parameters. Indicators of goodness of fit arise due to the difference between the empirical and theoretical covariance matrix [95].

Iacobucci [96] conducted a simulation study to examine the effect of sample size on goodness-of-fit measures. The study observed that sample size should be more than 50 (the fit measures are better when sample size increases). In addition, maximum likelihood estimation performed better because it was relatively robust to the multivariate normality

assumption. ML has been found to be relatively robust in case of violations of the multivariate normality assumption [97,98]. However, since there are no outliers in the sample under study, we did not expect non-normality to be a problem and we relied on the goodness of fit of the model. Moreover, "if the variables are reliable and the effects are strong and the model is not overly complex, smaller samples will suffice (Bearden, Sharma & Teel 1982; Bollen, 1990) " [96] (p. 91).

On the other hand, there are PLS (partial least squares) models that are SEMs based on the variance matrix (VB-SEM). However, PLS-SEM objective is to build theory based on the emerging relationships (in the same way as multiple regression analysis). Thus, it is a more exploratory model. PLS-SEM emerges from the discrepancy between the observed values and/or the dependent latent variables and the values estimated by the model [95]. Hence, measures or indicators of the predictive capacity of the model using PLS-SEM are of main importance to determine the quality or goodness of fit of the model, such as reliability and explained variance, among others, which is not needed in CB-SEM models [99]. Figure 1 shows the causal relationships to be tested.

*Variables*

The structural model is compound of two structural equations since there are two endogenous variables: SC activities and ICT use.

SC activities: SC activities is a variable (non-observable) obtained by using a factorial analysis among a set of indicators (Table 1).

**Table 1.** Communalities from factor analysis among SC activities. Source: own elaboration.

| Smart-City Dimension | SC Activities | Initial | First Extraction | Second Extraction |
|---|---|---|---|---|
| Smart environment | Reduction of private transports (car, moto) | 1.000 | 0.597 | 0.607 |
| | Use of public transport | 1.000 | 0.698 | 0.699 |
| | Use of bike lanes | 1.000 | 0.391 | |
| | Recycle or classify waste | 1.000 | 0.606 | 0.760 |
| | Rational water consumption | 1.000 | 0.723 | 0.742 |
| Smart government | Interaction with local government through internet (e-government) | 1.000 | 0.515 | 0.497 |
| Smart economy | Electronic commerce (sale-buy through internet) | 1.000 | 0.495 | 0.509 |
| Smart mobility | Use of SAMPEM Parking (parking app) | 1.000 | 0.562 | 0.586 |
| Smart people | Home-banking or financial transaction online (i.e.,: pay taxes) | 1.000 | 0.738 | 0.739 |
| | E-learning | 1.000 | 0.617 | 0.652 |

To develop a construct or variable for measuring SC activities of citizens from Bahía Blanca, a set of items was included based on a thorough review of the smart-city literature and mainly on the project's questionnaire. Citizens were asked to provide information about the kind of smart activities performed, from reduction of public transport to e-learning.

For instance, the survey explicitly asked if they performed any of the following activities to turn Bahía Banca into a smarter city (multiple choice): Reduction of private transports (car, moto); Use of public transport; Use of bike lanes; Recycle or classify waste; Rational water consumption; Interaction with local government through internet (e-government); Electronic commerce (sale-buy through internet); Use of SAMPEM Parking (parking app); Home-banking or financial transaction online (i.e.: pay taxes); and E-learning.

Both the Kaiser–Meyer–Olkin measure of sampling adequacy and Bartlett's test of sphericity indicate that the factor analysis may be useful with the data. The KMO measure of sampling was nearly 0.6; if the value were less than 0.50, the results of the factor analysis probably would not be very useful. On the other hand, the *p*-value related to Bartlett's test

of sphericity was zero, indicating that these variables were related and therefore suitable for structure detection.

Table 1 shows communalities from each SC activity acquired by performing a component principal analysis. We observed that the item or variable "Use of bike lane" was not relevant for explaining the variance (since communality from the first extraction was lower than 0.4). Then, we made the second extraction to define the significant items. After that, by using the components extracted, we built the SC activities index. Table 2 reports the first four factors that explained nearly 64 percent of the variance.

**Table 2.** Total variance explained.

| Components | Initial Eigenvalues | | | Extraction Sums of Squared Loadings | | | Rotation Sums of Squared Loadings | | |
|---|---|---|---|---|---|---|---|---|---|
| | Total | % of Variance | % Accumulated | Total | % of Variance | % Accumulated | Total | % of Variance | % Acummulated |
| 1 | 2.021 | 22.459 | 22.459 | 2.021 | 22.459 | 22.459 | 1.919 | 21.321 | 21.321 |
| 2 | 1.521 | 16.903 | 39.362 | 1.521 | 16.903 | 39.362 | 1.609 | 17.874 | 39.195 |
| 3 | 1.222 | 13.579 | 52.941 | 1.222 | 13.579 | 52.941 | 1.166 | 12.956 | 52.152 |
| 4 | 1.026 | 11.403 | 64.343 | 1.026 | 11.403 | 64.343 | 1.097 | 12.192 | 64.343 |
| 5 | 0.847 | 9.417 | 73.760 | | | | | | |
| 6 | 0.754 | 8.375 | 82.135 | | | | | | |
| 7 | 0.662 | 7.358 | 89.493 | | | | | | |
| 8 | 0.530 | 5.890 | 95.383 | | | | | | |
| 9 | 0.415 | 4.617 | 100.000 | | | | | | |

Extraction method: principal components analysis.

To construct the SC activities index, we computed the weighted sum of the four components. Each weight represented the proportion of variance of each component related to total variance. The average value of the SC activities index reached 0.51.

ICT use: This variable represents an independent explanatory factor of SC activities. The hypothesis was that the higher the ICT use, the higher the propensity of developing SC activities such as e-commerce and e-government would be.

ICT use is also an endogenous variable in the model, which is mainly explained by demographic factors. We used the same methodology as the SC activities index to build the variable ICT use. ICT use is a variable obtained from a factorial analysis among a set of ICT-use indicators. People were asked to provide information about the places used for connecting to the Internet as well as the devices employed. The higher the number of places and devices used for connecting to the Internet, the higher the intensity of using Internet will be.

Since we built ICT use as an exogenous variable by using factor analysis, we showed the suitability of the data for structure detection. Both the Kaiser–Meyer–Olkin measure of sampling adequacy (nearly 0.52) and Bartlett's test of sphericity (*p*-value 0.000) indicated that the factor analysis may be useful with the data. Tables 3 and 4 show information needed for this task.

**Table 3.** Communalities from factor analysis in ICT use. Source: own elaboration.

| | Initial | First Extraction | Second Extraction |
|---|---|---|---|
| Internet access at home | 1.000 | 0.862 | 0.868 |
| Internet access at work | 1.000 | 0.581 | 0.702 |
| Internet access in educational places | 1.000 | 0.572 | 0.571 |
| Internet access in commercial places | 1.000 | 0.625 | 0.670 |
| Internet access in public places | 1.000 | 0.701 | 0.736 |
| Use computer to connect Internet | 1.000 | 0.656 | 0.689 |
| Use mobile phones to connect Internet | 1.000 | 0.870 | 0.872 |
| Use tablet to connect Internet | 1.000 | 0.363 | |
| Use TV to connect Internet | 1.000 | 0.449 | 0.468 |
| Use e-Reader to connect Internet | 1.000 | 0.487 | 0.478 |

**Table 4.** Total variance explained. Source: own elaboration.

| Component | Initial Eigenvalues | | | Extraction Sums of Squared Loadings | | | Rotation Sums of Squared Loadings | | |
|---|---|---|---|---|---|---|---|---|---|
| | Total | % of Variance | Accumulated% | Total | % of Variance | Accumulated% | Total | % of Variance | Accumulated% |
| 1 | 1.948 | 21.646 | 21.646 | 1.948 | 21.646 | 21.646 | 1.714 | 19.048 | 19.048 |
| 2 | 1.684 | 18.715 | 40.362 | 1.684 | 18.715 | 40.362 | 1.651 | 18.347 | 37.395 |
| 3 | 1.302 | 14.469 | 54.830 | 1.302 | 14.469 | 54.830 | 1.382 | 15.357 | 52.752 |
| 4 | 1.120 | 12.446 | 67.277 | 1.120 | 12.446 | 67.277 | 1.307 | 14.524 | 67.277 |
| 5 | 0.936 | 10.398 | 77.675 | | | | | | |
| 6 | 0.751 | 8.339 | 86.014 | | | | | | |
| 7 | 0.571 | 6.340 | 92.354 | | | | | | |
| 8 | 0.434 | 4.824 | 97.178 | | | | | | |
| 9 | 0.254 | 2.822 | 100.000 | | | | | | |

Extraction method: principal components analysis.

Independent variables of SC activities:

Web-Trust: Trust in the web is an important factor of the willingness of citizens to use e-commerce and e-government, among other SC activities. Web trust refers to trust in the municipality's website; it is an ordinal Likert-scaled variable that ranges from 1 (totally disagree) to 5 (totally agree).

SC-Awareness: It is related to how much local citizens are aware of the smart-city concept. The survey asked each citizen if they were acquainted with the smart-city concept. It is an ordinal variable that ranges from "have never heard about that" to "I have heard about it but I do not know much", "I know the concept, what it means and how cities are implementing it", "I am researching about smart cities ideas" and, lastly, "I am planning or implementing one or more smart cities projects".

Independent variables of ICT use:

Age: Age is a scalar and continuous variable. Each participant was asked to report his or her age in the survey.

Education: High_education is a binary variable that takes value of 1 if the individual has reached a tertiary or university degree and zero otherwise (if the maximum education level is lower than tertiary).

Income: Permanent_job is a binary variable that takes the value of 1 if the individual works in a permanent job and zero otherwise. The survey did not collect information on income. However, labor condition can be a proxy of income. We supposed that citizens without a permanent job were in a more vulnerable condition to support connectivity and, therefore, to use ICT than the rest (people with transitory or informal jobs or unemployed people).

Gender: A binary variable that takes value of one if the individual is a man and zero if she is a woman.

## 4. Results

This SEM does not include a measurement model since there are no latent variables. Due to the limited number of observations, we did not add latent variables but included non-observable variables (such as ICT use and SC activities) by using factorial analysis.

In this section, we test the goodness of fit of the structural model (Table 5). The chi-square test is a test of exact fit that confirms that the structural equation model estimated can explain the theoretical model. Chi-square values near zero show adequate goodness of fit. Based on this model, the chi-square is 6.4 with a *p*-value of 0.38. As a result, the hypothesis of a perfect correspondence between the estimated matrix and the observations matrix cannot be rejected.

**Table 5.** Goodness of fit of the structural model. Source: own elaboration based on Stata 14.

| Statistic | Value | Description |
|:---:|:---:|:---:|
| Likelihood Ratio | | |
| Chi square_ms(6) | 6.405 | model vs. saturated |
| p > chi square | 0.379 | |
| Chi square_bs(13) | 23.281 | baseline vs. saturated |
| p > chi square | 0.028 | |
| Population Error | | |
| RMSEA | 0.027 | root mean square error of approximation |
| 90% CI, lowerbound | 0.000 | |
| upperbound | 0.139 | |
| Pclose | 0.623 | RMSEA probability <= 0.05 |
| Size of residuals SRMR | 0.038 | standardized root mean square residual |

On the other hand, we analyzed the root mean square error of approximation (RMSEA) to provide a more sensible approach than the chi-square test. The model fit is good if the RMSEA is lower than 0.05 and does not surpass the lower bound of the 90 percent confidence interval [100]. We could determine that the model fits well because the RMSEA reached a value of zero.

Moreover, there is an overall measure of bad fit known as the root mean square residual index (RMR). This indicator is based on the fitted residuals that are scale-dependent [101]. Therefore, the standard RMR (SRMR) is a more accurate indicator. If the SRMR is lower than 0.05, it indicates a good fit of the model [97].

Once the goodness of fit of the model was determined, we examined the structural model results (Table 6). Based on Table 6, ICT use positively affects SC activities ($\beta$ = 0.286) (hypothesis 1). Higher levels of ICT use show that citizens are more digitalized and probably have better digital skills than the rest. As the ICT use is greater, citizens can perform smart-city activities. People with lower ICT use will not be able to develop all kinds of smart cities. For instance, citizens without smartphones are less likely to use parking apps or use e-commerce.

**Table 6.** Structural models. Notes: ns means not significant; ** significant at 5% level. Source: own elaboration.

| Variable | Structural Equation SC Activities | Structural Equation ICT Use |
|:---:|:---:|:---:|
| ICT use | 0.2865806 ** | |
| Trust | 0.0063282 ns | |
| Awareness | 0.0492644 ** | |
| Education | | −0.0204209 ns |
| Labor_condition | | 0.075872 ** |
| Age | | −0.0029159 ** |
| Gender | | 0.0039698 ns |

In addition, awareness of the smart-city concept is a significant predictor of performing smart-city activities (hypothesis 2, $\beta$ = 0.049). The more aware people are of the smart-city concept, the more likely they are to perform smart activities. This result confirms previous literature on the subject [6,48]. Furthermore, trust in the website does not affect SC activities (hypothesis 3). Trust does not significantly explain SC activities. Therefore, we cannot confirm that citizens with a higher level of trust in the website are more likely to buy products and contact local governments. This finding follows Novo Vázquez and Vicente [32] but contradicts most of the literature [6,68].

On the other hand, labor condition and age are two significant factors of ICT use. Citizens with a permanent occupation show greater ICT use than the rest. In addition, the younger the citizen, the more likely they are to use ICT. This result follows previ-

ous studies [6,82]. Hence, young citizens with a permanent job will indirectly perform smart activities.

Moreover, the education level is not a significant factor of ICT use, contrary to the literature review provided [52,73,74]. This result is not surprising considering the sample selection bias. It is likely that respondents of an online survey have more digital skills and education than the whole population. On the other hand, gender is also not a significant factor of ICT use or SC activities.

## 5. Conclusions

The main contribution of this paper is offering some insights into citizens' engagement with smart cities in a city of a developing country, Argentina. In particular, the paper examines the determinants of smart-city activities across individuals from Bahía Blanca, Argentina. Although there is some empirical literature on this subject in developed countries, much less research has been conducted in the developing world where infrastructure and technological conditions are different. The paper also provides some insights useful for city stakeholders and policy makers to properly design smart-city policies and strategies for citizen engagement.

Based on the results obtained, citizens with different sociodemographic characteristics use ICT differently, and as a result, they perform smart activities in a different way, which supports the theoretical framework based on the UTAUT [12]. With respect to practical implications, local governments should focus their attention on performing ex ante promotion of ICT use and bringing awareness to the smart-city concept. Leading to smart cities implies reducing the digital gap in terms of ICT use. Local policies conducted to universalize Wi-Fi and Internet connectivity are not sufficient to amplify ICT use. In addition, a strategy of increasing ICT use would be useful to encourage smart-city activities or initiatives especially those related to environmental issues. Results show the environmental dimension has more weight among smart-city activities in Bahía Blanca. However, some activities are more developed in terms of the smart-city concept than others. For instance, use of public transport is more ICT-assisted than the rest. There is an app that shows real-time bus route and schedule information. On the contrary, there is a lack of digital technologies for a more effective waste management regime.

Moreover, since the smart-city concept is citizen-centric, a strategy to collect citizens' opinions and surveys about the degree of awareness of the smart-city concept should be promoted. In this vein, social media occupies a central role as a tool of communication and interaction between citizens (especially young citizens and those having a permanent job) and local government. Furthermore, this policy to increase smart-city awareness seems to be more efficient than motivating citizens to trust the web which resulted in a non-significant factor for encouraging SC activities. Public authorities should coordinate and cooperate with civic associations to raise trust and facilitate the understanding of smart-city initiatives and projects. This policy follows the international successful experience of global cities such as London, where citizens drive change in the same way as technocrats, businesses or policy makers.

Some limitations of the model are the omission of some variables such as citizens' political interests or political engagement, a sample size that is not large enough and sample selection bias since the survey was conducted on followers of the municipality's social media. Therefore, the results represent that population and not all citizens of Bahía Blanca. With respect to sample size, there is a consensus that larger samples are better than smaller ones. However, in the context of CB-SEM with a simple model to be estimated, non-measurement constructs (latent variables) and no presence of outliers, we are not concerned about the distribution of data. However, a larger sample would be desirable in case the population fraction of interest (the p parameter in the sample size formula) was larger than 50%. Finally, it would be useful to reproduce the survey with other citizen profiles, for instance, conducting a survey on citizens without Internet access or use.

**Funding:** This research was funded by Secretaría General de Ciencia y Tecnología, Universidad Nacional del Sur (UNS), Bahía Blanca, Argentina.

**Institutional Review Board Statement:** Not applicable.

**Informed Consent Statement:** Not applicable.

**Data Availability Statement:** The data presented in this study are available on request from the corresponding author.

**Conflicts of Interest:** The authors declare no conflict of interest.

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
