# Peer review of "Determinants of Smart City Commitment among Citizens from a Middle City in Argentina"

_smartcities, doi:10.3390/smartcities4030059_

Round 1

Reviewer 1 Report

Dear Author, the article is very interesting and important. I would like to present some comments. I hope you find them useful.

Substantive comments:

  1. in point 2.1. of the theoretical part of the study the scope of activity has been very narrowly defined, focusing on environmental issues
  2. in point 2.3. it is worth to add information in which countries Neirotti et al. conducted research (in the context of the explanation formulated in lines 182 and 183)
  3. in the theoretical justification of hypothesis 5, it is worth to explain more detailed the differences in activity between women and men
  4. the theoretical part can be extended by discussion of the studied dimensions of the smart city category (and possible activities)
  5. the study lacks at least a short characterization of the sample in terms of the variables studied; it would also be good to present the city itself in the context of its advancement in implementing the smart city idea
  6. you can supplement information about the website "Modernizia Baha" - what is its goal, who develops it and how, who observes it, etc.; are the respondents active on this website?
  7. the description of the content of the questionnaire requires clarification (line 268, scope of questions about "smart cicites", "e-government")
  8. it is worth expanding the discussion of the research results, indicating the conclusions in more detail, taking into account the obtained research results; conclusions should be formulated in relation to the activity of the examined city in the field of smart city implementation

Technical comments

  1. the structure of 2nd and 3rd part of the article could be improved; e.g. in Part 2, points 2.1. and subsequent ones appear after a long fragment of the text, which should be titled and numbered (2.1.)
  2. the "Results" point should be 4
  3. it is worth to use the same table standard, to clarify the column headings; maybe in tables 2 and 4 it is worth to use abbreviated component names, to clarify the title of the tables (e.g. 2 and 4)
  4. is the line 415 referring to table 5 or table 6?
  5. the sixth sentence ending in line 165 requires clarification - what kind of effectiveness and efficiency are meant
  6. incomprehensible fragment of the text in lines 82-84

Thank you. 

Author Response

Thank you for the comments, I hope the changes made have enhanced the understanding of the paper.

Below each comment I give you an answer. Yellow-marked text corresponds to changes in the manuscript. Green marked represents changes of order.

Dear Author, the article is very interesting and important. I would like to present some comments. I hope you find them useful.

Substantive comments:

in point 2.1. of the theoretical part of the study the scope of activity has been very narrowly defined, focusing on environmental issues

Answer: I add references and text to amplify the scope of the concept

in point 2.3. it is worth to add information in which countries Neirotti et al. conducted research (in the context of the explanation formulated in lines 182 and 183)

Answer: I add the information

in the theoretical justification of hypothesis 5, it is worth to explain more detailed the differences in activity between women and men

Answer: I add new references in the subject and explained briefly the results in the papers cited.

the theoretical part can be extended by discussion of the studied dimensions of the smart city category (and possible activities)

Answer: I add references to section 2.2.

the study lacks at least a short characterization of the sample in terms of the variables studied; it would also be good to present the city itself in the context of its advancement in implementing the smart city idea

Answer: I include a paragraph that briefly characterizes the sample profile in the methodology.

Also, iIn the introduction, I describe the city in terms of some indicators, and add a reference.

you can supplement information about the website "Modernizia Baha" - what is its goal, who develops it and how, who observes it, etc.; are the respondents active on this website?

Answer: I add some information about the page.

the description of the content of the questionnaire requires clarification (line 268, scope of questions about "smart cicites", "e-government")

Answer: I explicitly explain the question in the survey. I also add some paragraphs in theoretical framework about the different dimensions of a smart city

it is worth expanding the discussion of the research results, indicating the conclusions in more detail, taking into account the obtained research results; conclusions should be formulated in relation to the activity of the examined city in the field of smart city implementation

Technical comments

the structure of 2nd and 3rd part of the article could be improved; e.g. in Part 2, points 2.1. and subsequent ones appear after a long fragment of the text, which should be titled and numbered (2.1.)

Answer: I change the order of some paragraphs and add others.

the "Results" point should be 4

Answer: ok

it is worth to use the same table standard, to clarify the column headings; maybe in tables 2 and 4 it is worth to use abbreviated component names, to clarify the title of the tables (e.g. 2 and 4)

is the line 415 referring to table 5 or table 6? Answer: OK it is 6

the sixth sentence ending in line 165 requires clarification - what kind of effectiveness and efficiency are meant

Answer: it was corrected

incomprehensible fragment of the text in lines 82-84

Thank you.

Reviewer 2 Report

Smartcities-1259890

This is an important topic and an interesting approach to the issue of citizen participation in Smart Cities.  I had some trouble following the manuscript and I have a few questions about the research methodology.  At the end of the  comments  I have a few issues that are probably typos.

In general

I think that the paper needs to be reorganized to make the focus tighter.  As it stands now,  it’s difficult to follow your argument.  I would suggest (1) use a social science framework such as APA and (2) have someone edit the paper for you.

A structural equation model seems like more analysis than you need for a small accidental sample.  I also wonder if it's appropriate given the way you drew the sample.  There are parametric tests.

Are there 97 cases in the site  (in which case that’s the population, rather than a sample) or you received responses from 97? 

You need to tie your conclusions to your discussion and reference your review of the literature.

Minor Issues

First Page—There is something missing here:

 There are many 33 examples of smart city services such as end-to-end applications that focus on common concerns such as mobility, public services, and safety. Recently, this crucial role has been  evidenced because of the pandemic crisis (Kunzmann, 2020).

Second Page

Innovation Diffusion Theory    Diffusion of Innovation Theory???

Author Response

Thank you for the comments, I hope the changes made have enhanced the understanding of the paper.

Below each comment I give you an answer. Yellow-marked text corresponds to changes in the manuscript. Green marked represents changes of order.

This is an important topic and an interesting approach to the issue of citizen participation in Smart Cities.  I had some trouble following the manuscript and I have a few questions about the research methodology.  At the end of the  comments  I have a few issues that are probably typos.

In general

I think that the paper needs to be reorganized to make the focus tighter.  As it stands now,  it’s difficult to follow your argument.  I would suggest (1) use a social science framework such as APA and (2) have someone edit the paper for you.

A structural equation model seems like more analysis than you need for a small accidental sample.  I also wonder if it's appropriate given the way you drew the sample.  There are parametric tests.

Answer: The idea of using SEM was to build a model with a moderator variable as it is ICT use. Thus, ICT use (which is centered in the citizen) plays a key role in the relationship between demographic factors ans SC engagement.

Are there 97 cases in the site  (in which case that’s the population, rather than a sample) or you received responses from 97?

Answer: In methodology I explain the population and sample. 97 is the sample, the population is the 16th thousand followers of the city’s social media.

You need to tie your conclusions to your discussion and reference your review of the literature.

Answer: I think the discussion of results is complete. In each hypothesis I compare with the references from the theoretical framework. I also add some recommendations in conclusions.

Minor Issues

First Page—There is something missing here:

 There are many 33 examples of smart city services such as end-to-end applications that focus on common concerns such as mobility, public services, and safety. Recently, this crucial role has been  evidenced because of the pandemic crisis (Kunzmann, 2020).

Answer: OK, changed

Second Page

Innovation Diffusion Theory    Diffusion of Innovation Theory???

Changed

Round 2

Reviewer 2 Report

Smartcities-1259890   Revision

Thank you for your revisions. The paper is stronger but problems remain.

  • While the focus is better, it is still difficult to follow. Some of the additions to the conclusions are difficult to understand.  A good edit might be helpful.
  • The research question and importance of the study should be in the beginning.
  • The sample is at best a purposive sample at worst a convenience or accidental sample. The analysis that you are using requires a multivariant normal distribution.  This doesn’t appear to be what you have here.You need to explain why this is appropriate.  You also need to discuss this as limitations of the study.
  • Can we assume that the response rate was 100% and there were only 97 people in the population?
  • Speaking of limitations of the study, these need to be included in the discussion
  • Let's take this out “We examined the city of Bahía Blanca because is the place of residence of the author”. Everybody knows that.  We all do that.

Author Response

Thank you for the comments. Below each comment is my answer. 

  • While the focus is better, it is still difficult to follow. Some of the additions to the conclusions are difficult to understand.  A good edit might be helpful.

Answer: I corrected the english writing of the conclusions.

  • The research question and importance of the study should be in the beginning.

Answer: I add a research question to the introduction

  • The sample is at best a purposive sample at worst a convenience or accidental sample. The analysis that you are using requires a multivariant normal distribution.  This doesn’t appear to be what you have here.You need to explain why this is appropriate.  You also need to discuss this as limitations of the study.

Answer: I add references that support the sample size taking into account it is a ML estimation based on CB-SEM.

In methodology I wrote:

Iacobucci (2010) conducted a simulation study to examine the effect of sample size on fit measures. The study observed that sample size should be more than 50 (the goodness of fit measures are better when sample size increases). Besides, the Maximum Likelihood estimation performed better because it was relatively robust to the multivariate normality assumption. ML has been found to be relatively robust in case of violations of the multivariate normality assumption (Hu & Bentler, 1995; Olsson et al., 2000). However, since there are no outliers in the sample under study, we do not expect non-normality to be a problem and we rely on the goodness of fit of the model. Moreover, “if the variables are reliable and the effects are strong and the model not overly complex, smaller samples will suffice (Bearden, Sharma & Teel 1982; Bollen, 1990) (Iacobucci (2010), p.91).

Besides, I add some additional considerations in conclusions and methodology

Informal answer: Certainly, it was not a convenience or a purposive sample. By the time of sharing the survey on the social media, we had expected a larger number of participants. However, taking into consideration the low level of engagement of citizens, the size becomes reasonable. On the contrary, I would not believe in a survey with a larger number of answers with this methodology and in this area of research.

  • Can we assume that the response rate was 100% and there were only 97 people in the population?
  • Answer: I add a footnote in methodology about that:The original sample (N=99) included two invalid cases of individuals who did not live in the city. As a result, there were too many missing data in those cases. These two cases were not included in the analysis.

  • Speaking of limitations of the study, these need to be included in the discussion
  • Answer: The last paragraph in conclusions is about limitations.

  • Let's take this out “We examined the city of Bahía Blanca because is the place of residence of the author”. Everybody knows that.  We all do that.
  • OK
